# Characterization Study of Empty Fruit Bunch (EFB) Fibers Reinforcement in Poly(Butylene) Succinate (PBS)/Starch/Glycerol Composite Sheet

**DOI:** 10.3390/polym12071571

**Published:** 2020-07-15

**Authors:** Rafiqah S. Ayu, Abdan Khalina, Ahmad Saffian Harmaen, Khairul Zaman, Tawakkal Isma, Qiuyun Liu, R. A. Ilyas, Ching Hao Lee

**Affiliations:** 1Laboratory of Biocomposite Technology, INTROP, Universiti Putra Malaysia, Serdang 43400, Selangor, Malaysia; ayu.rafiqah@yahoo.com (R.S.A.); harmaen@upm.edu.my (A.S.H.); 2Engineering Faculty, UPM, Serdang 43400, Selangor, Malaysia; intanamin@upm.edu.my; 3Polycomposite Sdn Bhd, Jalan Maharajalela, Hilir Perak 36000, Perak, Malaysia; dr.khairulz@gmail.com; 4The BioComposites Centre, Bangor University, Bangor LL57 2UW, UK; q.liu@bangor.ac.uk; 5Advanced Engineering Materials and Composites Research Centre (AEMC), Department of Mechanical and Manufacturing Engineering, Universiti Putra Malaysia, Serdang 43400, Selangor, Malaysia; ahmadilyasrushdan@yahoo.com

**Keywords:** empty fruit bunch fiber (EFB), polybutylene succinate (PBS), starch, glycerol, characterizations, biocomposite, polymer Blends

## Abstract

In this study, a mixture of thermoplastic polybutylene succinate (PBS), tapioca starch, glycerol and empty fruit bunch fiber was prepared by a melt compounding method using an industrial extruder. Generally, insertion of starch/glycerol has provided better strength performance, but worse thermal and water uptake to all specimens. The effect of fiber loading on mechanical, morphological, thermal and physical properties was studied in focus. Low interfacial bonding between fiber and matrix revealed a poor mechanical performance. However, higher fiber loadings have improved the strength values. This is because fibers regulate good load transfer mechanisms, as confirmed from SEM micrographs. Tensile and flexural strengths have increased 6.0% and 12.2%, respectively, for 20 wt% empty fruit bunch (EFB) fiber reinforcements. There was a slightly higher mass loss for early stage thermal decomposition, whereas regardless of EFB contents, insignificant changes on decomposition temperature were recorded. A higher lignin constituent in the composite (for high natural fiber volume) resulted in a higher mass residue, which would turn into char at high temperature. This observation indirectly proves the dimensional integrity of the composite. However, as expected, with higher EFB fiber contents in the composite, higher values in both the moisture uptake and moisture loss analyses were found. The hydroxyl groups in the EFB absorbed water moisture through formation of hydrogen bonding.

## 1. Introduction

The development of biodegradable materials has attracted much research interest by scientists on worldwide. Aliphatic polyesters are among the most promising materials for the production of high-performance biodegradable plastics. One of the polyesters, polybutylene succinate (PBS) which is commercially available in the market, has very high fame as a high-performed bioplastic [1]. Many recent studies have selected PBS as the composite matrix for various applications and purposes [2,3,4].

PBS is synthesized from succinic acid and 1,4-butanediol (BDO) via a polycondensation process, and exhibits balanced performance in thermal and mechanical properties as well as processability [5]. It is more thermally stable than PLA polymer [6]. PBS is able to undergo biodegradation and even disposal in compost, moist soil, fresh water (by activated sludge), or sea water. It also can be composted by microorganism activities to convert it into CO_2_, H_2_O, and inorganic products under aerobic conditions, or CH_4_, CO_2_, and inorganic products under anaerobic conditions. The biodegradability of PBS depends mainly on its chemical structure and especially on its hydrolysable ester bond in the main chain, which is susceptible to microbial attack [7,8]. One study prepared a reactive-PBS polymer (RPBS) with insertion of toluene-2,4,diisocyanate (TDI) chemical in different ratios and blends with starch. The properties of the blended specimens were found to be significantly improved, even with only 10 wt% of RPBS. The TDI chemical insertion smoothened the PBS/starch polymer blend’s surface, showing better miscibility of the two phases [9]. However, PBS has some negative properties such as slow crystallization rate, low melt viscosity, and softness. These have restricted its processing condition and potential applications. Polymer mixing with other materials is commonly used, to develop new blend materials that are suitable for specific working environments or specific purposes. However, most of the polymers are not miscible with each other and tend to phase-separate in a melt state [10]. Besides, although a fast crystallization reaction can happen when mixing with other materials, this may cause deterioration of PBS composite’s strength [11]. Therefore, plasticizers such as glycerol were added to overcome and improve the flexibility of PBS polymer [12]. The council of the IUPAC (International Union of Pure and Applied Chemistry) has defined a plasticizer as ‘‘a substance or material incorporated in a material (usually a plastic or elastomer) to increase its flexibility, and workability by lowering glass transition temperature (T_g_)” [13]. Glycerol is a pure anhydrous structure and has a specific gravity of 1.261 g·mL^−1^, melting point of 18.2 °C and boiling point of 290 °C under normal atmospheric pressure [14]. On the other hand, grafting is another method to improve a compatibilizer between two materials. Suchao-in et al., 2013, have grafted PBS on tapioca starch blends. Results revealed a strong interfacial adhesion of the blend and enhanced modulus properties, as evidenced from SEM micrographs [15].

Starch is one of the materials that is readily available, low cost and one of the important bioresources used in the food industry, e.g., as a thickener and gelling agent. It also possesses good physical, mechanical and oxygen barrier properties, that give it potential to become active film [6,16]. It is much more reliable and chemically stable than other spacers [17]. Starch is a natural polymeric product and is found in almost every plant. Usually the main sources of starch come from tapioca, potato, maize, rice and wheat [18]. Starch contains two different molecular structures, linear (1,4)-linked α-d-glucan amylose and highly (1,6)-branched α-d-glucan amylopectin. The starch molecules are tied by van der Waals bonds and strong intermolecular hydrogen bonds. Common native starch granules have a semi-crystalline, radially oriented spherulitic structure. They contain water on different structural levels [19]. Amylopectin consist of a branching chain that forms double helices and produce crystalline structure of the granules, whereas amylose is amorphous and interspersed among amylopectin molecules [20]. Some starch polymers form helical structures due to the existence of α linkages, which contribute to its extraordinary properties and enzyme digestibility [21]. The relative amounts of amylose and amylopectin depend upon the plant source. Corn starch granules typically contain approximately 70% amylopectin and 30% amylose [22]. However, native starch itself cannot be satisfactorily used due to its hydrophilicity and brittleness which lead to the poor mechanical properties, so it requires some chemical modification to overcome this drawback [23]. Blending thermoplastic starch with PBS is one of the frequently selected options by researchers. Higher water resistance, good processability, fully biodegradable, and superior mechanical properties were being claimed for PBS/corn starch blend with glycerol plasticizers [24].

On the other hand, extensive investigation has been carried out to study the effects of natural fiber reinforcement on polymer composites [25,26,27]. The majority of outcomes have agreed that reinforced natural fiber has a better performing load transfer mechanism, and results in higher mechanical properties [28,29]. Empty fruit bunch (EFB) fibers have shown comparable quality to high strength kenaf bast fibers [30]. However, the hydrophilic nature of the EFB fiber is found to be incompatible with the hydrophobic polymer matrix. This caused poor interfacial adhesion between the fiber and matrix, leading to lower performances. Chemically treated EFB fibers had greater thermal and morphologies properties [31]. Moreover, it consists of wood-like constituents (cellulose, hemicellulose and lignin), showing lower thermal stability towards high heat environments, yet producing high residue at high temperature [32]. Furthermore, the hydrophilic behavior is expected to have higher moisture absorption, leading to swelling of the EFB fiber. Nevertheless, the extremely low cost of EFB fiber as a byproduct and its 100% biodegradable properties have created a high interest in it [33].

This study is a continuation of previous study, which investigated the characterization of high volume contents of EFB fiber reinforced in PBS/tapioca starch composite [34]. The high volume of fiber reinforcement found deterioration of mechanical properties due to poor interfacial bonding, evidenced from SEM micrograph and this is not accepted by the market, and similar findings were reported that show a lower tensile strength when alkaline treated-sugarcane fibers were inserted without any plasticizers [35]. Hence, in the present study, a lower volume of EFB fiber was added into the PBS/starch composite sheet with glycerol plasticizers to improve compatibility. This study has filled the knowledge-of-gap on low EFB fiber reinforcement in PBS/starch composite sheet with plasticizer fillers. The outcomes of this investigation (mechanical, morphological and thermal characterization) could serve as valuable knowledge for future developments on EFB fiber reinforcement in polymer composite.

## 2. Experimental

### 2.1. Materials

PBS in the form of pallets were bought from PTT Public Company Limited in Thailand. Density of PBS is 1.26 g/cm^3^. Tapioca starch in form of powder was obtained from PT Starch solution in Indonesia. Empty fruit bunch fiber (EFB) was used and obtained from Polycomposite Sdn Bhd in Negeri Sembilan. The EFB were chopped using a grinder machine and sieved to get an average 300–600 µ in size. Meanwhile, glycerol was purchased from Duro Kimia Sdn Bhd in Selangor. The properties of materials as tabulated in Table 1.

### 2.2. PBS Composite Preparation

The PBS pallets and EFB fiber was first dried in an oven at 80 °C to prevent excessive hydrolysis which can compromise physical properties of the polymer. Starch, glycerol and EFB were dry mixed in an industrial mixer machine and sieved to remove excessive lumps during the mixing process. Then, PBS and the mixed compound of starch/EFB/glycerol were added into an industrial counter rotating extruder feeder for a total of 300 kg per processing. After that, the compound was melted in an industrial extruder machine comprising 10 heat zones, which were set temperatures in between 115–145 °C with rotation speed of 80 RPM. As a result of the shear stress imposed on fibers during compounding, homogenization of PBS/starch/fiber/glycerol was carried out by cycling the mixture in the extruder for 15 min and then extruded through a 2 mm gauge strand die at a rate of 10 mm/s. The melted compound was then passed through a calendaring machine before producing a sheet. Then, the sheets were cut into shapes according to specific characterization testing. The image of the extruded compound is shown in Figure 1.

#### 2.3.1. Mechanical Properties (Tensile Properties)

The tensile testing of the composite was conducted using a 5 kN Bluehill INSTRON Universal Testing Machine. The test was carried out according to ASTM standard D-638. The specimens were cut into dog bone shape by a plastic molder machine with the specifications of 120 × 120 × 2 mm^3^ of length, width and thickness respectively. The composites were gripped at a 30 mm gauge length and the crosshead speed was set at 2.0 mm/min. All specimens were kept in a conditioning room and the test was run at 22 °C and relative humidity (RH) at 55%. Seven specimens were tested per test condition.

#### 2.3.2. Mechanical Properties (Flexural Properties)

Flexural test of the composite was performed using 5 kN Bluehill INSTRON Universal Testing Machine. Test samples were cut to the dimension of 70 × 15 × 2 mm^3^ and three-point bending tests were performed according to ASTM D790 standard. The crosshead speed was set at 2 mm/min with a support span-to-depth ratio of 16:1. All specimens were kept in a conditioning room and the test was run at 22 °C with the relative humidity (RH) at 55%. Seven specimens were tested per test condition.

#### 2.3.3. Morphological Analysis

Morphology of the samples was observed using Hitachi S-3400N scanning electron microscope (SEM) equipped with energy dispersive X-ray (EDX) under an accelerating voltage of 15 kV and at an emission current of 58 µA. The tensile-tested-samples were gold sputtered before observation to avoid the charging effect during sample examination. SEM helps to analyze the microscopic structure and characterization of the compound on the basis morphology and structural changes.

#### 2.3.4. Thermal Analysis

The thermal stability of the samples was characterized using a TA Instruments Q500 thermogravimetric analyzer, TGA. About 6 mg of the sample was scanned from 30 to 700 °C at a heating rate of 20 °C min^−1^ under a nitrogen gas atmosphere.

#### 2.3.5. Moisture Absorption and Moisture Loss Analysis

Sample sheets of rectangular shape with dimensions of 15 × 15 × 0.5 mm^3^ were dried in a vacuum oven at 60 °C for 24 h and weighed prior to testing. The vacuum dried rectangular sheets were immersed in distilled water at 20 °C to determine the water absorption and soluble ratio. The sample was taken out to measure the water absorption and soluble ratio in a certain time, and then the same sample was vacuum dried to measure the weight loss of the sample. The weights of the original sample and the sample after water absorption were designated as *W*_0_ and *W*_1_, and the dry weight of the water extracted sample was designated as *W*_2_. The value of moisture absorption was obtained by Equation (1):(1)$$\mathrm{Moisture}\;\mathrm{uptake}=\frac{W_{1}-W_{2}}{W_{2}}\times 100\%$$
with the value of the soluble ratio derived from Equation (2):(2)$$\mathrm{Soluble}\;\mathrm{ratio}=\frac{W_{0}-W_{2}}{W_{0}}\times 100\%$$

Three measurements were performed for each sample, and the result was reported as the average value. This procedure followed the short-term immersion standard method ASTM D570-98.

On the other hand, seven samples were prepared for the moisture content evaluation. The samples were placed in normal climatic conditions at room temperature (27 ± 2 °C) with 65% relative humidity of air for 24 h before being weighed. Percentages of moisture content were determined by using Equation (3). The samples were heated in the oven for 24 h at 105 °C. Before heating the samples were measured as *M*_0_. After 24 h in the oven, the fiber was weighed again as *M*_1_. Therefore:(3)$$\mathrm{Moisture}\;\mathrm{content}\;(\%)\;\frac{M_{1}-M_{0}}{M_{0}}\times 100\%\;$$

## 3. Result and Discussion

### 3.1. Mechanical Testing

Filler reinforcement is an important factor in determining mechanical properties of the composite. The most crucial factor that affects the mechanical properties of the fiber reinforced materials is its fiber/matrix interfacial adhesion. The strength of the interfacial bonding was determined by several factors, such as the nature of the fiber and polymer components, fiber aspect ratio and processing procedure [36,37]. The mechanical properties of the PBS composite are presented and illustrated in Table 2 and Figure 2 and Figure 3, respectively. It was clearly shown that the tensile and flexural strength of specimens were decreased for fiber reinforcement up to 8 wt%. This is due to poor dispersion and incompatibility between fillers and the PBS matrix according to previous studies [38,39]. Fibers are unable to disperse evenly in the PBS matrix, creating high stress concentration spots, resulting in a dramatic reduction in tensile strength [40]. However, increments in tensile and flexural strength were observed, indicating that the reinforcing ability of the natural fibers has overcome the shortage from the interfacial adhesion factor. A previous study reported the same trend, that higher fiber contents led to an improvement in the tensile strength of the matrix due to the interaction related to the fiber contents [41].

On the contrary, there were relatively higher mechanical properties for a 0% EFB specimen (which contained 30 wt% of starch/glycerol with a 2:1 ratio) in a current study, when compared to a previous study, which only gave 16.12 and 21.78 MPa for tensile and flexural strength, respectively, for pure PBS polymer [34]. The insertion of starch supposedly reduces the composite’s strength performance due to low compatibility [6]. However, the addition of glycerol has the adverse effect of strength deterioration by localization of a compatibilizer at the interface for a stable morphology from a SEM micrographic [42].

Accoding to Thirmizir et al., the flexural strength of PBS composites was higher than neat PBS polymer. [8]. Higher fiber loadings have improved the flexural strength due to mechanical interlocks found between fiber and matrix. The fiber/matrix mechanical interlocking was expected to act as a mechanism to withstand the bending force in flexural testing. On the other hand, flexural strength was reduced by 6% for 8 wt% EFB fiber reinforcement composites. This may be attributed to interruption of the continuous long polymer chain by the presence of hydrophilic lignocellulose. Similarly, higher flexural strength values were recorded for higher EFB fiber reinforcement specimens. EFB fibers work as a carrier of loads in the matrix, synchronized with the tensile performance.

### 3.2. Morphological Analysis

Figure 4 shows SEM images for specimens’ surface morphology, under 500× magnification. The strength performances of the composites are directly affected by morphology status. Figure 4a shows a smooth and regular PBS surface, while Figure 4b shows the image of modified tapioca starch granules on the surface. Figure 4c,d shows the presence of EFB fiber, which consists of long fibers surrounded on the PBS/starch matrix. The poor adhesion of fibers on the matrix shows correlation with the reduction of mechanical properties for “8 wt% EFB” specimens. The poor impregnation makes it easier for the fiber to be pulled out, and causes a lower strength performance for the composite. This trend was also reported by a previous researcher [36]. For Figure 4e,f, it can be observed that the fibers are adhered to the matrix. The longitudinal fibrous shapes of the fibers were evenly mixed and evenly distributed on the matrix surface. The fibers mix homogenously with the matrix and are not clearly seen on the surface morphology analysis. This indicates the good fiber/matrix adhesion. On the other hand, the void between the EFB fiber and matrix is less, which gives a better fiber/matrix bonding and increased mechanical strength to the composite.

### 3.3. Thermal Analysis

Thermogravimetric analysis (TGA) is a useful method for quantitative determination of the degradation behavior, thermal stability and mass change in a composite. The appearance of starch and glycerol in the PBS polymer composite has reduced the thermal stability of the specimen in generally. Excess amounts of glycerol have taken part in the reaction with hydroxyl groups of the PBS polymer, which promoted a lower thermal stability [43]. However, with more starch/glycerol contents replaced by EFB fibers, the effects of glycerol are lesser and gradually dominated by EFB fibers.

Figure 5 shows the TGA profiles of the EFB composites, while Table 3 lists the mass loss in every stage with peak temperature until sample reach 600 °C. There is a small but noticeable step between 75–95 °C, which was due to the presence of free water in the composite. Other researchers also have reported that this is due to water removal, as starch has a higher tendency to absorb moisture [6]. It also was reported that at the initial stage weight loss may be ascribed to the evaporation of water in the fiber [32,44,45]. Sharp transitions at peak 2 and 3 between 200–265 °C is due to decomposition of polysaccharide components in the starches. At higher temperatures, hemicellulose degradation occurs, followed by cellulose degradation [46]. Both degradation processes involve complex reactions (dehydration, decarboxylation, among others) as well as breakage of C-H, CO and C-C bonds [47]. Apart from this, lignin starts to degrade at a temperature range between 250–450 °C. Lignin degradation generates water, methanol, carbon monoxide and carbon dioxide [48,49]. PBS matrix is a thermally stable biopolymer and it begins to degrade near 300 °C with high degradation rates, as similarly found by Lee et al. [50]. In this analysis, there was slightly higher mass loss for early stage thermal decomposition whereas insignificant changes on decomposition temperature, regardless of EFB contents. However, with the higher lignin constituent in the composite there was a higher mass residue, which would turn into char at high temperature. This observation indirectly proves the dimensional integrity of the composite. Besides, the better mechanical performance for the high natural fiber reinforcement could offer wider the applications for this composite material.

### 3.4. Moisture Uptake and Average Loss of Moisture Contents

The amount of water absorbed in the composite was calculated by weight difference between before and after samples exposed to water. Figure 6 shows moisture uptake over the time and average loss of moisture contents for EFB composites. The moisture uptake test was conducted to identify the amount of water absorbed by the composites while the average loss of moisture content is to measure the mass loss after being subjected to heat. Generally, the moisture uptake was depending on several factors such as volume fraction of fiber, voids, viscosity of matrix, humidity and temperature [51].

Water absorption is one of the disadvantages of applying lignocellulosic materials. Insertion of starch components into PBS polymer (0% EFB), comes with expected higher water absorption [52,53], as the starch component may take up to 300% of water absorption, as reported previously [9]. However, when a portion of the starch/glycerol is replaced by EFB fiber, higher values are found in both moisture uptake and loss analyses. This is because of the hydrophilic properties of the natural fibers in the poor interfacial bonding, leading to higher increments of moisture uptake, due to the presence of hydroxyl groups. Hence, it was observed that 20% EFB composite has the highest moisture uptake. The hydroxyl groups absorbed water moisture through formation of hydrogen bonding. The higher moisture content of the natural fiber may result in a weak interfacial bonding between the fiber and matrix [54]. The water molecules were absorbed in the inter-fibrillar space of the cellulosic structure that exists in the fiber and causes cracks and micro voids in the composite surface [55]. During immersion of the samples in water, capillarity action conducts water molecules to fill the voids, causing cracks and dimensional change. Swelling of fiber also leads to interfacial debonding and thereby reduction of mechanical strength [56,57]. In this study, at 6 to 8 h immersion, samples reached stable moisture contents, showing a saturation point, where no more water was absorbed. Similarly, when subjected to heat, the high EFB loadings composite loses a higher amount of water content. This shows that fiber reinforcement improves strength profiles, yet may cause higher susceptibility to moisture attack, thereby reducing overall composite properties.

## 4. Conclusions

In this study, the effect of fiber content on the mechanical and thermal properties of polybutylene succinate (PBS) composites were mainly evaluated. The control specimen (0% EFB) was compared with PBS polymer to discuss the changes affected by the appearance of starch/glycerol components. Generally, insertion of starch/glycerol provided better strength performance, but worse thermal and water uptake to all specimens.

On the other hand, it was found that there was poor interfacial adhesion between the EFB and PBS matrix, leading to lower mechanical properties. Fortunately, this was overcome and improved by higher fiber reinforcement, that regulated a better load transfer mechanism. Higher fiber loadings have improved the flexural strength due to mechanical interlocks found between the fiber and matrix. As a result, the tensile and flexural strength had increases of 6.0% and 12.2%, respectively, for 20 wt% EFB reinforcements.

In the SEM micrographic, it shows a smooth surface for PBS, while appearances of the EFB fiber show poor adhesion on the matrix, and was found to correlate with the mechanical properties analysis. On the other hand, the void between the EFB fiber and matrix was less and gave better fiber/matrix for a high fiber volume content composite.

A total of four thermal degradation peaks were recorded in the TGA analysis. The first peak was observed at 75–95 °C, due to the presence of free water in the composite. Sharp transitions at peak 2 and 3 between 200–265 °C were due to decomposition of the polysaccharide components in the starches and natural fibers. The last thermal decomposition peak was recorded at around 350 °C, which was responsible for the degradation of the PBS matrix. In this analysis, there was a slightly higher mass loss for early stage thermal decomposition, whereas insignificant changes on decomposition temperature were recorded, regardless of EFB contents. However, the higher lignin constituent in the composite had a higher mass residue, which would turn into char at high temperature. This observation indirectly proves the dimensional integrity of the composite. Moreover, the better mechanical performance of the high natural fiber reinforcement could offer wider applications for this composite material.

The moisture uptake over time and average loss of moisture contents for EFB composites were analyzed in this study. The higher the EFB fiber content in the composite, the higher values in both moisture uptake and loss data were found. This is expected due to the hydrophilic properties of the natural fibers that lead to higher increments of moisture uptake, due to the presence of hydroxyl groups. Hence, it was observed that 20% EFB composite has the highest moisture uptake. In this study, at 6 to 8 h immersion, samples reached a stable moisture content, showing a saturation point, where no more water was absorbed. Similarly, when subjected to heat, the high EFB loadings composite loses a higher amount of water content. This shows that fiber reinforcement improves the strength profile yet may cause higher susceptibility to moisture attack, thereby reducing overall composite properties.

As concluding remarks, the present results suggest that the use of 20% EFB fiber contents in the composite may be a potential candidate for effectively improving the properties and performances of the composite for future application. Nevertheless, the content of starch/glycerol may need to strategically planned to obtain a balance between performance and costing.

## Figures and Tables

**Figure 1 polymers-12-01571-f001:**
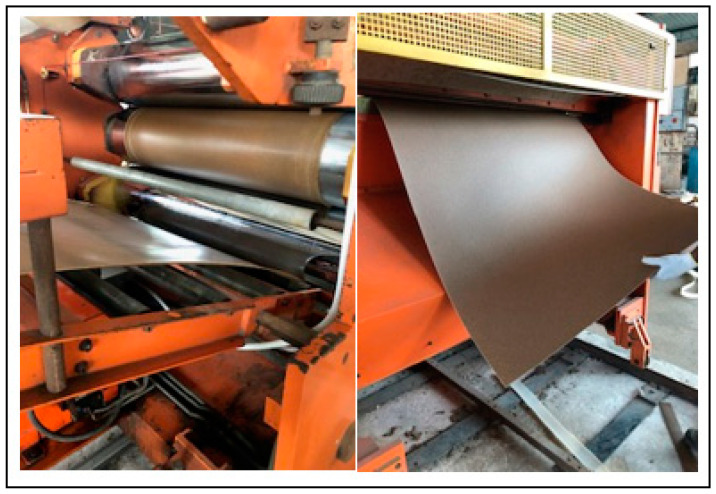
Sheet extrusion process.

**Figure 2 polymers-12-01571-f002:**
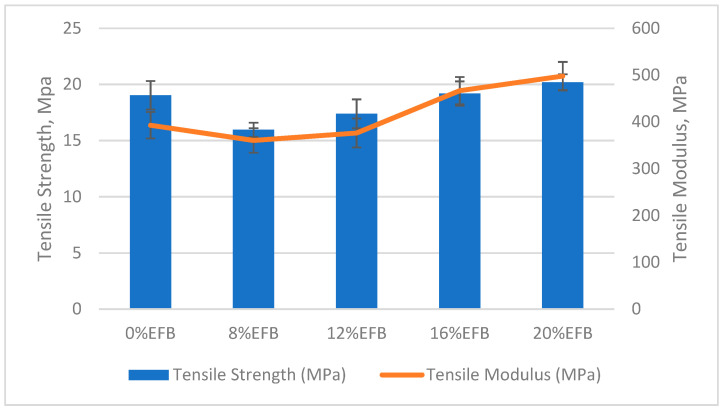
Tensile strength and tensile modulus of PBS composites.

**Figure 3 polymers-12-01571-f003:**
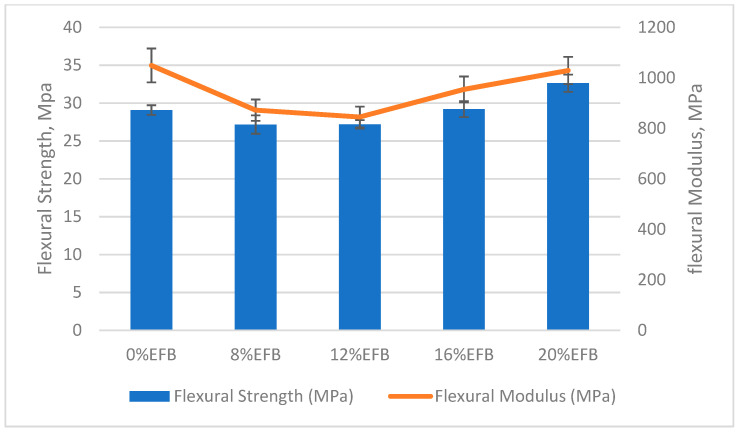
Flexural strength and flexural modulus of PBS composites.

**Figure 4 polymers-12-01571-f004:**
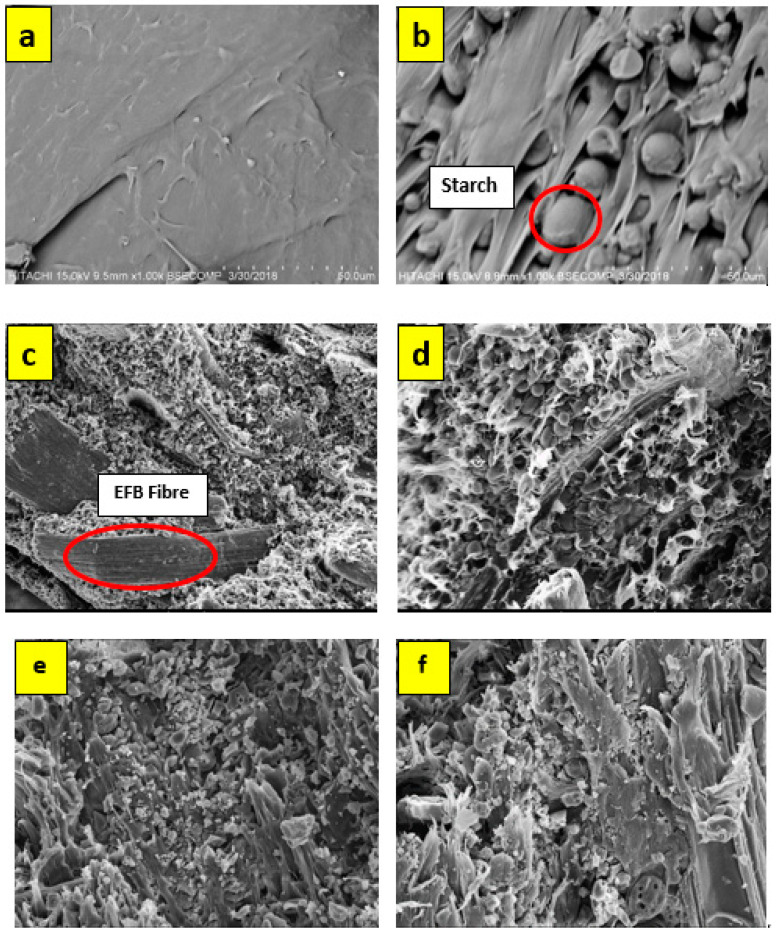
SEM micrographs for: (**a**) raw PBS, (**b**) 0% EFB, (**c**) 8% EFB, (**d**) 12% EFB, (**e**) 16% EFB and (**f**) 20% EFB.

**Figure 5 polymers-12-01571-f005:**
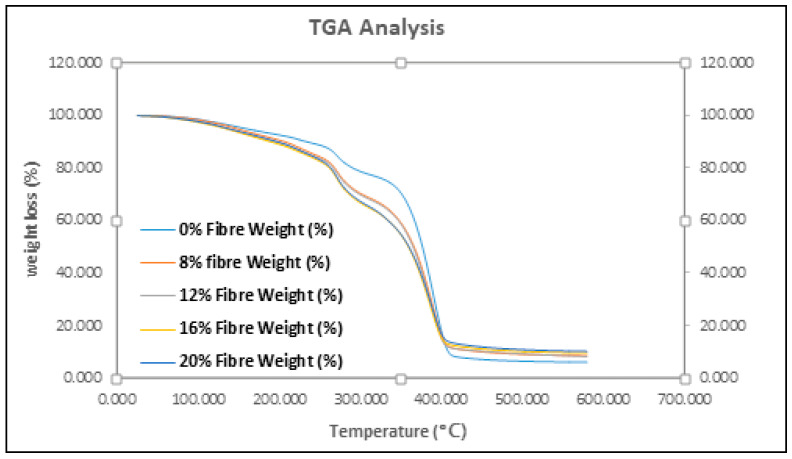
TGA profiles of EFB composites.

**Figure 6 polymers-12-01571-f006:**
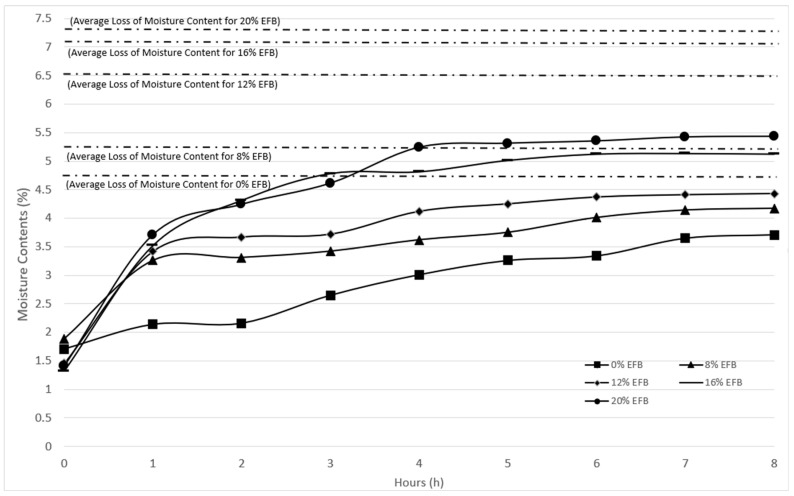
Moisture uptake analysis and average loss of moisture contents for EFB composites.

**Table 1 polymers-12-01571-t001:** Properties of polybutylene succinate (PBS), starch and empty fruit bunch (EFB) fiber.

Properties	PBS	Starch	Properties	EFB Fiber
Density (g/cm^3^)	1.26 g/cm^3^	0.63	Density	0.98 g/cm^3^
MFR	5 g/10 min	None	Cellulose (%)	45
Color	White	White	Lignin (%)	23
Odor	No Odor	No Odor	Hemicellulose (%)	21
Melting Point	115 °C	None	Size Mesh (µ)	300–600
Molecular Weight	65,000 g/mol	692.7 g/mol	Moisture (%)	9.41

**Table 2 polymers-12-01571-t002:** Mechanical properties of PBS/starch/glycerol and EFB blends.

Formulation ^a^	Specimen	Tensile Strength (MPa)	Tensile Modulus (MPa)	Flexural Strength (MPa)	Flexural Modulus (MPa)
PBS 70% Starch/Glycerol 30%	0% EFB	19.04 ± 1.27	392.76 ± 28.16	29.08 ± 0.64	1049.13 ± 67.15
PBS 70%, EFB 8%, Starch/Glycerol 22%	8% EFB	15.96 ± 0.63	360.40 ± 26.17	27.17 ± 1.21	872.10 ± 42.36
PBS 70%, EFB 12%, Starch/Glycerol 18%	12% EFB	17.38 ± 1.29	376.33 ± 31.06	27.19 ± 0.56	845.17 ± 41.17
PBS 70%, EFB 16%, Starch/Glycerol 14%	16% EFB	19.19 ± 1.08	466.84 ± 29.14	29.20 ± 1.05	954.35 ± 51.25
PBS 70%, EFB 20%, Starch/Glycerol 10%	20% EFB	20.18 ± 0.72	497.95 ± 30.17	32.63 ± 1.14	1029.15 ± 54.15

^a^ Starch/Glycerol in 2:1 ratio.

**Table 3 polymers-12-01571-t003:** A summary of peak temperatures for EFB composites.

Specimens	Peak 1, °C	Mass loss, %	Peak 2, °C	Mass loss, %	Peak 3, °C	Mass loss, %	Peak 4, °C	Mass loss, %	Mass Residue, %
0% EFB	78.13	7.161	214.45	3.404	261.43	12.19	362.05	71.22	5.995
8% EFB	95.13	8.086	209.19	6.774	262.82	16.63	358.92	59.95	8.492
12% EFB	87.89	9.534	209.61	6.129	262.12	16.42	358.85	59.63	8.176
16% EFB	88.50	9.885	-	-	254.79	25.07	357.29	55.38	9.568
20% EFB	84.87	5.940	201.68	6.634	260.63	21.82	357.93	55.32	10.16

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
