# Peer review of "Characterization Study of Empty Fruit Bunch (EFB) Fibers Reinforcement in Poly(Butylene) Succinate (PBS)/Starch/Glycerol Composite Sheet"

_polymers, 2020, doi:10.3390/polym12071571_

Round 1
Reviewer 1 Report
In view of the overlap with your article published in Scientific Reports ("Effect of Empty Fruit Brunch reinforcement in PolyButylene-Succinate/Modified Tapioca Starch blend for Agricultural Mulch Films", https://www.nature.com/articles/s41598-020-58278-y), and with articles 10.1016/j.carbpol.2010.08.051 and 10.1016/j.polymertesting.2018.02.003 (which were not cited in the text), one can conclude that the novelty is very limited. In my opinion, it does not meet the novelty and significance requirements to deserve publication in Polymers.
Author Response
First of all, we would like to show our appreciation to reviewers, for handling this manuscript. We shall explain the whole plan of the project. This paper is under a sponsored project which exploring and widening the use of empty fruit bunch fibres (EFB) in biocomposites. The clarification between previous and current papers has been made in the manuscript (in the last paragraph of the introduction).
In the previous paper published in Scientific Reports, high volume content of fibres (30-50 wt%) has been applied. However, the mechanical properties of EFB biocomposite were not satisfying. Hence, in the current paper, we applied a lower amount of EFB fibres reinforcement (0-18 wt%) yet including glycerol plasticizer with the purpose of improving polymer flexibility. Therefore, the present study is a continuing study of the previous investigation and it has distinct differences between two studies. More importantly, the properties difference between high- and low-volume of EFB reinforcement shall be compared and provide valuable knowledge to all readers for future EFB fibre reinforcement development.
I hope the explanation above did help in clarifying the differences and novelty of present study.
Reviewer 2 Report
This research described the inclusion of EFB fiber in PBS matrix and how the mechanical and thermal properties can be changed.
- the EFB fiber did not increase the mechanical properties that much, especially at lower EFB content. This is contradictory to many reports when the concentration is low and the dispersion quality is high, reinforcement can be efficient. What is the reason?
- what is the interactions between EFB fiber and the matrix? Can interaction parameters be quantified, e.g., via dynamic mechanical analysis?
Author Response
Reply to Question 1
Thank you for your valuable opinion. In the analysis of fibre insertion, effective reinforcement can be achieved by numerous factors as well as deterioration of properties.
For a low volume of fibre reinforcement, the well-dispersion of fibres did improve the mechanical properties of the composite. However, poor interfacial bonding between hydrophilic fibre/ hydrophobic matrix, improper fabrication parameters (high temperature, prolonged mixing period etc) leading to fibre damage, high void contents, inhomogeneous mixing, and a lot more factors may compensate the good side of fibre insertion and resulted in lower mechanical properties.
The below has listed some of the previous studies using low fibre volume and resulted in lower tensile strength.
- Rozman, H. D., Lim, P. P., Abusamah, A., Kumar, R. N., Ismail, H., & Ishak, Z. A. M. (1999). The Physical Properties of Oil Palm Empty Fruit Bunch (EFB) Composites Made from Various Thermoplastics. International Journal of Polymeric Materials, 44(1-2), 179–195
- Ngo, W.L. & Pang, Ming & Yong, Lengchuan & Tshai, Kim Yeow. (2014). Mechanical Properties of Natural Fibre (Kenaf, Oil Palm Empty Fruit Bunch) Reinforced Polymer Composites. Advances in Environmental Biology. 8. 2742-2747.
Reply to Question 2
We have shown the interactions between EFB fibre and matrix by using SEM micrographic. Poor interfacial bonding between fibre/matrix but well dispersion indicates the status of interaction. However, there is no standard testing that can quantify the degree of interaction.
Dynamic mechanical analysis (DMA) is one of the useful thermal analysis. It shows the heat absorb/release behaviour of the composite. However, interaction parameters could not identity by DMA testing.
Round 2
Reviewer 1 Report
If, as indicated in previous iteration, "[...] the present study is a continuing study of the previous investigation and it has distinct differences between two studies. More importantly, the properties difference between high- and low-volume of EFB reinforcement shall be compared and provide valuable knowledge to all readers for future EFB fibre reinforcement development. [...]", such differences and the claimed novelty should not only be highlighted in a single paragraph in the introduction, but in a detailed discussion. In the revised document, I have only found a reference to the Scientific Reports paper ([31]) in the introduction. The rest of the document has not been improved.
Please expand the discussion section to clearly show how the research reported herein can contribute to the body of knowledge, commenting in detail the differences with your previous paper, with the two suggested references (10.1016/j.carbpol.2010.08.051 and 10.1016/j.polymertesting.2018.02.003) or with other relevant references that you may find after a thorough bibliographical survey. In its current form, reasonable doubts still arise regarding the significance and novelty of the study. Please kindly note that I have not co-authored aforementioned two references, so there is no conflict of interest (I am explicitly asking to provide a comparison with them because, in my view, they clearly affect the novelty of this article).
Author Response
- If, as indicated in previous iteration, "[...] the present study is a continuing study of the previous investigation and it has distinct differences between two studies. More importantly, the properties difference between high- and low-volume of EFB reinforcement shall be compared and provide valuable knowledge to all readers for future EFB fibre reinforcement development. [...]", such differences and the claimed novelty should not only be highlighted in a single paragraph in the introduction, but in a detailed discussion. In the revised document, I have only found a reference to the Scientific Reports paper ([31]) in the introduction. The rest of the document has not been improved
We had gone through every sentence in manuscript and revised the manuscript to highlight and discuss the novelty of current manuscript.
- Please expand the discussion section to clearly show how the research reported herein can contribute to the body of knowledge, commenting in detail the differences with your previous paper, with the two suggested references (10.1016/j.carbpol.2010.08.051 and 10.1016/j.polymertesting.2018.02.003) or with other relevant references that you may find after a thorough bibliographical survey. In its current form, reasonable doubts still arise regarding the significance and novelty of the study. Please kindly note that I have not co-authored aforementioned two references, so there is no conflict of interest (I am explicitly asking to provide a comparison with them because, in my view, they clearly affect the novelty of this article)
To highlight the novelty and difference compared to previous studies. Some general discussions on glycerol insertion have been made in the discussion section. In the past study, there is no research investigation on the EFB fiber/PBS/Starch/glycerol composite.
Therefore, we're looking on title revise to,
Characterization Study of Empty Fruit Bunch (EFB) Fibers Reinforcement in Poly(Butylene) Succinate (PBS)/Starch/Glycerol Composite Sheet
Reviewer 2 Report
accept
Author Response
Thank you very much for your time and acceptance of this manuscript.
Round 3
Reviewer 1 Report
No further changes are needed, as far as I am concerned. Thank you for addressing the queries raised in previous iterations.